# TOWARDS ROBUST AND REALISTIC HUMAN POSE ESTIMATION VIA WIFI SIGNALS

## ABSTRACT

Robust WiFi-based human pose estimation (HPE) is a challenging task that bridges discrete and subtle WiFi signals to human skeletons. We revisit this problem and reveal two critical yet overlooked issues: 1) cross-domain gap, i.e., due to significant discrepancies in pose distributions between source and target domains; and 2) structural fidelity gap, i.e., predicted skeletal poses manifest distorted topology, usually with misplaced joints and disproportionate bone lengths. This paper fills these gaps by reformulating the task into a novel two-phase framework dubbed ***DT-Pose***: **D**omain-consistent representation learning and **T**opology-constrained ***Pose*** decoding. Concretely, we first propose a temporal consistency contrastive learning strategy with uniformity regularization, integrated into a self-supervised masked pretraining paradigm. This design facilitates robust learning of domain-consistent and motion-discriminative WiFi representations while mitigating potential mode collapse caused by signal sparsity. Beyond this, we introduce an effective hybrid decoding architecture that incorporates explicit skeletal topology constraints. By compensating for the inherent absence of spatial priors in WiFi semantic vectors, the decoder enables structured modeling of both adjacent and overarching joint relationships, producing more realistic pose predictions. Extensive experiments conducted on various benchmark datasets highlight the superior performance of our method in tackling these fundamental challenges in 2D/3D WiFi-based HPE tasks. The code is available in the supplementary materials.

## 1 INTRODUCTION

Image-based human pose estimation (HPE), a highly active and hot topic, has recently achieved remarkable success in both 2D Cao et al. (2017); Wang et al. (2022) and 3D scenarios Li et al. (2022); Gong et al. (2023), spanning single-person Zhang et al. (2021) and multi-person settings Shi et al. (2022); Liu et al. (2023). These advancements have significantly propelled broad applications in virtual reality Zheng et al. (2023), autonomous driving Zheng et al. (2022), and the healthcare community He et al. (2024b). However, those visual-based methods face inherent limitations due to realistic challenges (e.g., lighting intensity, view variations, and occlusions). Furthermore, rising concerns regarding privacy have driven the growing research attention toward non-visual modalities (e.g., WiFi Yan et al. (2024); D Gian et al. (2025), RF Fan et al. (2025), and wearable sensor Chen et al. (2020) signals), which offer significant advantages in privacy protection and resilience to occlusions. Among these, the WiFi modality holds promise due to its widespread deployability and compatibility with Edge AI in the AIoT era.

Tracing the development of the WiFi-based HPE, the field has gradually progressed from single-person 2D to more complex multi-person 3D HPE Wang et al. (2019a;b); Jiang et al. (2020); Ren et al. (2022); Zhou et al. (2022); Yang et al. (2022); Ren et al. (2021); Zhou et al. (2023); Yan et al. (2024); D Gian et al. (2025). Predominantly, these methods rely on supervised learning and focus on designing complex regression networks to map WiFi signals to 2D/3D pose coordinates. However, all of them assume that the training and testing data follow the same distribution, which does not hold in real-world scenarios due to domain variability. To address this limitation, the recently introduced WiFi dataset (MM-Fi Yang et al. (2024)) incorporates cross-domain settings, presenting new challenges for evaluating the generalizability of WiFi-based HPE methods.

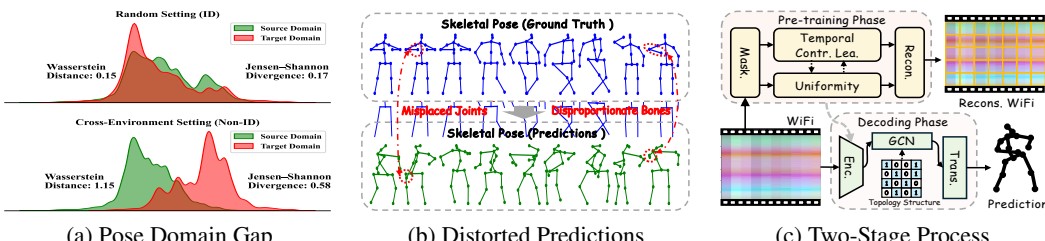

(a) Pose Domain Gap      (b) Distorted Predictions      (c) Two-Stage Process

Figure 1: (a) shows the pose coordinates distribution between the source and target domains. (b) represents the predictions of the MetaFi++ method Zhou et al. (2023) and corresponding ground truth. (c) denotes the overview framework of our method.

Upon analyzing WiFi signals across diverse domains, we observed significant discrepancies in pose coordinate distributions between the source and target domains under cross-environment settings, i.e., ***cross-domain gap***, contrasting with the commonly held assumption of identical distributions (Fig. 1 (a)). In such scenarios, existing supervised methods tend to overfit source-domain pose distributions and generalize poorly to target domains. This limitation underscores the inadequacy of supervised learning in capturing intrinsic motion patterns embedded in sparse WiFi signals, leading to the learning of spurious, motion-irrelevant, and noisy features. While AdaPose Zhou et al. (2024) also recognizes this challenge, its reliance on pre-acquired target domain data for domain adaptation renders it impractical and suboptimal. Thus, we aim to design a WiFi-specific approach that learns domain-consistent and motion-discriminative WiFi representations independent of pose coordinates, thereby enhancing cross-domain transferability.

In addition to domain generalization challenges, we observe that existing methods often produce pose predictions with unrealistic topologies (e.g., misplaced joints and disproportionate bone lengths), resulting in a ***structural fidelity gap*** (Fig. 1(b)). These deficiencies stem from two key factors: (1) prior works typically adopt CNNs combined with MLPs to regress poses in an unconstrained manner, leading to poor modeling of human joint relationships; and (2) unlike the image modality, which provides explicit spatial priors (e.g., human heatmaps), the WiFi modality offers only high-level semantic representations (e.g., global vectors) that lack spatial topological information, making it inherently more difficult to capture valid pose structures. To mitigate these issues, we propose incorporating explicit skeletal topology priors as constraints to better model the non-trivial spatial relationships among human joints.

Building on the above observations, we propose a novel framework (***DP-Pose***), which reformulates WiFi-based HPE as a two-phase process: ***D***omain-consistent WiFi representation learning and ***T***opology-constrained ***Pose*** decoding, as depicted in Fig. 1 (c). In the first phase, we transform raw WiFi signals into image-like inputs and adopt the self-supervised masked prediction strategy of MAE He et al. (2022) as the main line to learn domain-consistent representations. Considering the temporal continuity of WiFi signals, we treat adjacent WiFi frames within an action sequence as positive pairs and others in the batch as negatives, yielding motion-discriminative representations through contrastive objectives. Additionally, uniformity regularization is employed to mitigate potential representational collapse caused by signal sparsity. In the second phase, the pre-trained encoder is frozen to extract domain-consistent WiFi representations. We then introduce task prompts and Graph Convolution layers with spatial topology priors as constraints, enabling localized modeling of adjacent joint relationships. Concurrently, we establish more holistic dependencies among overarching joints within Transformer layers. By exploring these adjacent and overarching spatial correlations, our decoding architecture promotes realistic and structurally coherent pose predictions.

The main contributions can be summarized as follows:

- We reveal the cross-modal gap issue in WiFi-based HPE and develop a tailored WiFi representation learning method that integrates a temporal consistency contrastive strategy with uniformity regularization, enabling the extraction of domain-consistent and motion-discriminative features from sparse signals.

- We reveal the structural fidelity gap issue in WiFi-based HPE and propose a hybrid decoding architecture with explicitly incorporated skeletal topology priors as constraints, compensating for the lack of spatial cues in WiFi vectors and enabling effective modeling of joint relationships.

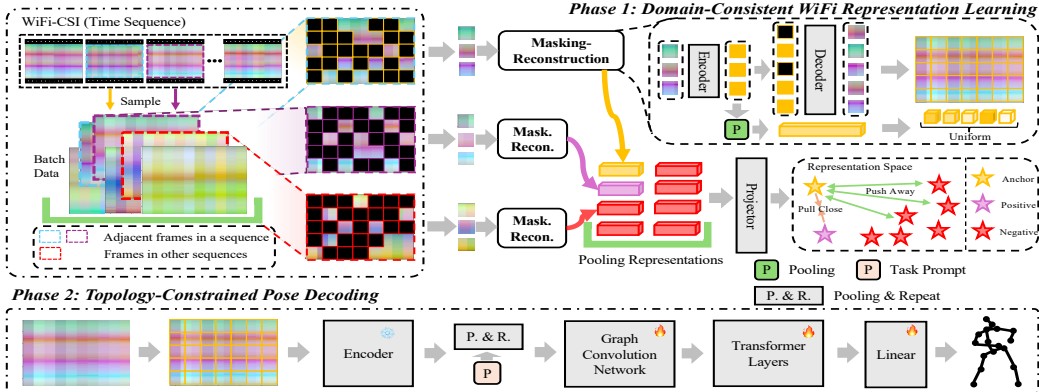

Figure 2: The pipeline of our method, including the pre-training and pose decoding phases (zoom in for a better view).

- We evaluate the effectiveness of our method through extensive and comprehensive experiments on three mainstream datasets, demonstrating its superior performance in the WiFi-based HPE field.

## 2 RELATED WORK

WiFi-based HPE is an emerging research topic that has gradually flourished in recent years, encompassing a range of tasks from single-person 2D Wang et al. (2019a); D Gian et al. (2025) and 3D Jiang et al. (2020); Ren et al. (2022; 2021); Zhou et al. (2023) to multi-person 2D Wang et al. (2019b) and 3D scenarios Yan et al. (2024). Early work in this field, such as WiSPPN Wang et al. (2019a;b), pioneers 2D HPE by employing fundamental CNN models He et al. (2016). Subsequently, WiPose Jiang et al. (2020) extends to 3D poses through a combination of CNN and RNN layers, thereby leveraging temporal dynamics to yield smoother skeletal predictions. Differently, both GoPose Ren et al. (2022) and Winect Ren et al. (2021) methods leverage the 2D angle-of-arrival features of WiFi signals to estimate 3D poses. Recently, such as MetaFi++ Zhou et al. (2023) and Person-in-WiFi-3D Yan et al. (2024), employ the Transformer layers to learn WiFi representations for single-/multi-person 3D HPE. Concurrently, HPE-Li D Gian et al. (2025) has designed dynamic CNN kernels to predict poses more efficiently. ***Unfortunately, all these studies overlook cross-domain gaps and rely on pose supervision in the source domain to guide WiFi representation learning from sparse signals, which hinders generalization to target domains with different pose distributions. Even more fundamentally, WiFi signals lack explicit human spatial priors by nature compared to images, making it challenging to perceive body topology when using the abovementioned CNN architectures directly.*** (More related works are provided in the Appendix A.) [***Summary***]: In this work, we are the first to tackle the cross-domain gap challenge by introducing a self-supervised pretraining strategy with WiFi-specific designs, tailored to the sparse and continuous nature of WiFi signals. Simultaneously, we explicitly capture both adjacent and overarching spatial correlations between joints by compensating skeletal topology priors into a hybrid decoding architecture, thereby ensuring structurally faithful pose predictions.

## 3 METHOD

### 3.1 PRELIMINARY

Typically, WiFi signals are captured using multiple transmitters and receivers, with each signal comprising multiple subcarriers operating in orthogonal frequency bands to facilitate inter-device communication. These subcarriers describe the signal propagation process, known technically as Channel State Information (CSI). As shown in Fig. 3 (a), the CSI undergoes various distortions attributed to multipath effects and physical transformations, e.g., reflections, diffraction, and scattering Yan et al. (2024); Zhou et al. (2023). Leveraging these properties, we can record time-continuous

WiFi signals that are dynamically influenced by human activities, i.e., action movements, thereby enabling the estimation of corresponding human poses. More specifically, one WiFi sample can be represented as $\mathbf{X} \in \mathbb{R}^{E \times R \times A \times S \times T}$, where $E, R, A, S$ denote the numbers of transmitters, receivers, antennas, and subcarriers, respectively. Here, $T = \frac{f_{\text{wifi}}}{f_{\text{video}}}$ represents the temporal resolution, equal to the ratio of the WiFi sampling frequency to that of the corresponding video action sequence. Notably, increasing the number of subcarriers and antennas enhances the resolution of the WiFi signals, capturing more subtle movements and finer variations. We define the ground truth of 3D pose coordinates for each frame as $\mathbf{Y} \in \mathbb{R}^{M \times J \times C}$, where $M$ represents the number of humans, $J$ indicates the number of joints, and $C$ specifies the spatial dimensions (coordinates). Hence, the entire dataset can be formalized as $\mathcal{D} = \{\mathbf{X}_i \in \mathbb{R}^{E \times R \times A \times S \times T}, \mathbf{Y}_i \in \mathbb{R}^{M \times J \times C}\}_{i=1}^N$, where $N$ is the total number of samples.

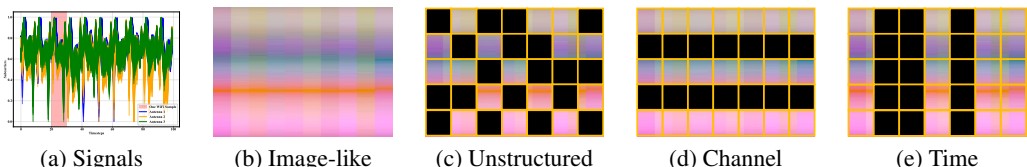

| (a) Signals | (b) Image-like | (c) Unstructured | (d) Channel | (e) Time |

Figure 3: Original WiFi CSI signals and different masking strategies on the MM-Fi dataset.

## 3.2 DOMAIN-CONSISTENT REPRESENTATION LEARNING

**Masked Operation.** To more closely align the WiFi modality with the image-based framework employed in MAE He et al. (2022), we first reshape each WiFi sample into an image-like form $\hat{\mathbf{X}}_i \in \mathbb{R}^{A \times ERS \times T}$, as illustrated in Fig. 3 (b). Concretely, we treat the antenna dimension $A$ as the image channels, the concatenated subcarriers from all devices ($ERS$) to image height, and the temporal resolution $T$ as image width. Subsequently, to investigate the best suitable masking strategy tailored to WiFi signals, we consider three distinct approaches at the pre-training stage, including unstructured (i.e., random masking with grid shape), channel-structured (i.e., random masking subcarriers along the time-frequency axis), time-structured (i.e., random masking timesteps along the subcarrier axis), as shown in Fig. 3 (c) - (e). Following MAE He et al. (2022), we divide $\hat{\mathbf{X}}_i$ into non-overlapped regular grid patches and employ convolution layers to embed each patch, obtaining $\tilde{\mathbf{X}}_i \in \mathbb{R}^{n \times d}$, where $n$ is the patch numbers and $d$ is the embedding dimension. We then incorporate fixed sinusoidal positional embeddings into these embedded patches and apply random masking with a high ratio (80% in our experiments) to enforce robust representation learning. The encoder, comprising 4 stacked Transformer layers, is tasked with learning domain-consistent WiFi representations, while 2 Transformer layers in the decoder strive to reconstruct the original WiFi $\hat{\mathbf{X}}_i$, ultimately producing a reconstruction $\mathbf{X}_i'$. The entire procedure is optimized by minimizing MSE between the reconstruction and the original input as follows:

$$\mathcal{L}_{\text{Mask}} = ||\mathbf{X}_i' - \hat{\mathbf{X}}_i||_2^2. \tag{1}$$

**Temporal Consistency Strategy.** Continuous WiFi samples captured over the duration of an action sequence can reflect the temporal variation of the motion. However, masked operation for individual WiFi samples would only extract modality-specific representations, lacking the essential motion patterns, as shown in Fig. 5 (a). Thus, we treat adjacent WiFi frames within the same action sequence as a positive pair due to motion consistency within them. Notice that the WiFi samples in a batch include one-pair adjacent WiFi frames $(\hat{\mathbf{X}}_t, \hat{\mathbf{X}}_{t+1})$ from the same sequence and other non-isomorphic WiFi frames $\{\hat{\mathbf{X}}_i\}_{i=1}^{\text{B}-2}$ from all action sequences, where B is the batch size. Thus, other combinations of WiFi samples in a batch should be negative pairs. Following the masked procedure, we pool the encoded visible embedding patches of each sample to derive a representation $\mathbf{e}_i \in \mathbb{R}^d$. Next, we project them to obtain positive pair representations $(\mathbf{s}_t, \mathbf{s}_{t+1})$ and other sample representations $\{\mathbf{v}_i\}_{i=1}^{\text{B}-2}$. Then, we pull the positive pair closer and push the negative pairs away in a batch based on InfoNCE loss as follows, where $\rho(\cdot)$ is the cosine similarity, $\phi(\cdot)$ is the $\exp(\cdot)$ function, and $\tau$ is the temperature parameter.

$$\mathcal{L}_{\text{CL}} = -\log \frac{\phi(\rho(\mathbf{s}_t, \mathbf{s}_{t+1})/\tau)}{\phi(\rho(\mathbf{s}_t, \mathbf{s}_{t+1})/\tau) + \sum_{i=1}^{\text{B}-2} \phi(\rho(\mathbf{s}_t, \mathbf{v}_i)/\tau)}. \tag{2}$$

**Objective with Uniformity Regularization.** The masked process and temporal consistency strategy jointly ensure the extraction of domain-agnostic and motion-discriminative WiFi representations. However, due to the inherent sparsity and homogeneity of WiFi signals, the learned representations may suffer from dimensional collapse, as depicted in Fig. 6. Here, we introduce explicit uniformity regularization to enhance representation diversity as follows:

$$\mathcal{L}_{\text{unif}} = \frac{1}{B} \sum_{j=1, j \neq i}^{B} (\hat{\mathbf{e}}_i^\top \hat{\mathbf{e}}_j)^2, \hat{\mathbf{e}}_i = \|\mathbf{e}_i\|_2, \hat{\mathbf{e}}_j = \|\mathbf{e}_j\|_2. \tag{3}$$

Overall, the pre-training optimization objective for *each WiFi sample* can be formulated as follows:

$$\mathcal{L} = \mathcal{L}_{\text{Mask}} + \lambda_{\text{CL}} \cdot \mathcal{L}_{\text{CL}} + \lambda_{\text{unif}} \cdot \mathcal{L}_{\text{unif}}, \tag{4}$$

where $\mathcal{L}_{\text{CL}}$ and $\lambda_{\text{unif}}$ are trade-off hyperparameters.

### 3.3 Topology-Constrained Pose Estimation

**Adjacent Joint Local Modeling.** After the pre-training phase, we freeze the pretrained encoder to extract WiFi representations $\mathbf{F} \in \mathbb{R}^{n \times d}$. To align these representations with the structure of human joints, we first pool all patches into one vector and repeat them into the joint numbers. Next, we add the learnable task prompt on them to obtain $\hat{\mathbf{F}} \in \mathbb{R}^{J \times d}$ for structural pose shape learning. Furthermore, we represent the human skeleton as a graph, where each joint is a vertex and each bone is an edge. This structure allows us to incorporate explicit spatial topology prior $\mathbf{A} \in \{0, 1\}^{J \times J}$, where $\mathbf{A}_{i,j} = 1$ indicates that the $i$-th joint and $j$-th joint are physical connected and $\mathbf{A}_{i,j} = 0$ otherwise. By employing Graph Convolution layers, we leverage $\mathbf{A}$ as a structural constraint to aggregate information from spatially connected joints, thereby mitigating the lack of spatial priors in $\hat{\mathbf{F}}$. Formally, the updated representation $\tilde{\mathbf{F}}$ is computed as follows:

$$\tilde{\mathbf{F}} = \sigma(\mathbf{D}^{-\frac{1}{2}} \mathbf{A} \mathbf{D}^{-\frac{1}{2}} \hat{\mathbf{F}} \mathbf{W}), \tag{5}$$

where $\mathbf{D} \in \mathbb{R}^{J \times J}$ is the degree matrix for normalization, $\mathbf{W}$ is a learnable parameter, $\sigma$ is the activate function.

**Overarching Joint Holistic Modeling.** Beyond local relationships, it is essential to capture holistic, long-range correlations among overarching joints. To this end, we treat the joints as an ordered sequence and apply Transformer encoder layers to enhance their non-physical interdependencies, such as the potential relationships between head and hand joints in "drinking water" pose. We calculate the attention values among all joints as follows:

$$\mathbf{Q} = \tilde{\mathbf{F}} \mathbf{W}_Q, \mathbf{K} = \tilde{\mathbf{F}} \mathbf{W}_K, \mathbf{V} = \tilde{\mathbf{F}} \mathbf{W}_V, \tag{6}$$

$$\mathbf{Z}_{\text{attn}} = \text{LN}(\tilde{\mathbf{F}} + \text{softmax}(\frac{\mathbf{Q} \mathbf{K}^T}{\sqrt{\tilde{d}}}) \mathbf{V}), \tag{7}$$

where $\mathbf{W}_Q, \mathbf{W}_K, \mathbf{W}_V$ are learnable parameters, $\tilde{d}$ is the dimension of $\mathbf{K}$, and $\text{LN}(\cdot)$ denotes the layer normalization. Then, we feed them into the feed-forward network $\text{FFN}(\cdot)$ and regress them into pose coordinates by MLPs $\Psi(\cdot)$:

$$\mathbf{Z} = \text{LN}(\text{FFN}(\mathbf{Z}_{\text{attn}}) + \mathbf{Z}_{\text{attn}}), \quad \hat{\mathbf{Y}}_i = \Psi(\mathbf{Z}), \tag{8}$$

where $\hat{\mathbf{Y}}_i$ is the predicted pose. By jointly capturing local and holistic dependencies, our hybrid decoder produces predicted structurally coherent and realistic poses.

**Objective.** For training the pose decoder, we adopt the MSE loss for *each sample* to regress the pose as follows:

$$\mathcal{L} = \|\hat{\mathbf{Y}}_i - \mathbf{Y}_i\|_2^2. \tag{9}$$

## 4 Experiments

To evaluate the effectiveness of DT-Pose, we conduct comprehensive experiments across three mainstream datasets: **MM-Fi** Yang et al. (2024), **WiPose** Zhou et al. (2022), and **Person-in-WiFi-3D** Yan et al. (2024). The dataset introduction is in Appendix B. For additional experimental details, results, and analyses beyond those presented below, we refer readers to the Appendix.

## 4.1 2D & 3D HPE PERFORMANCE COMPARISON

As shown in Table 1, 2, 3, and 7 (in Appendix), our framework outperforms existing methods in both 2D and 3D WiFi-based HPE tasks, illustrating its versatility and robust generalization. In particular, superior PA-MPJPE results highlight the plausibility of the predicted poses and the structural coherence of the generated skeletons. Notably, the remarkable gains under cross-domain settings in Table 1 confirm that our pretraining strategy successfully captures generalizable representations. Furthermore, our method also delivers competitive performance in terms of efficiency in Table 2.

Table 1: **3D HPE** results on **MM-Fi** dataset. Best and second-best are in **Red** and Blue, respectively. † indicates results reproduced from released code but not reported in the original paper. Results for **2D HPE** on **MM-Fi (P3-S1)** dataset are shown in Table 7 of Appendix.

| Method | Protocol 1 | | | | Protocol 2 | | | | Protocol 3 | | | |
|---|---|---|---|---|---|---|---|---|---|---|---|---|
| | PCK@20↑ | PCK@50↑ | MPJPE↓ | PA-MPJPE↓ | PCK@20↑ | PCK@50↑ | MPJPE↓ | PA-MPJPE↓ | PCK@20↑ | PCK@50↑ | MPJPE↓ | PA-MPJPE↓ |
| *Setting 1 (Random Split)*: | | | | | | | | | | | | |
| MetaFi++ Zhou et al. (2023) | 49.1† | 86.5† | 186.9 | 120.7 | 32.2† | 81.7† | 213.5 | 121.4 | 43.9† | 85.0† | 197.1 | 121.2 |
| HPE-Li D Gian et al. (2025) | 56.2† | 87.6† | 173.4† | 104.5† | 36.9† | 81.9† | 206.1† | 102.7† | 49.6† | 85.6† | 184.3† | 106.4† |
| **DT-Pose (Ours)** | **59.4** | **88.9** | **165.3** | **101.0** | **41.4** | **83.5** | **195.6** | **101.2** | **51.7** | **86.5** | **178.5** | **104.5** |
| *Setting 2 (Cross-Subject)*: | | | | | | | | | | | | |
| MetaFi++ Zhou et al. (2023) | 36.4† | 85.5† | 222.3 | 125.4 | 24.0† | 77.5† | 247.0 | 122.7 | 32.3† | 81.9† | 231.1 | 124.0 |
| HPE-Li D Gian et al. (2025) | 38.2† | 82.8† | 228.6† | 106.8† | 26.9† | 78.0† | 242.6† | 101.9† | 36.5† | 80.8† | 228.6† | 107.7† |
| **DT-Pose (Ours)** | **41.9** | **86.7** | **213.0** | **105.6** | **28.5** | **78.5** | **238.3** | **101.1** | **37.7** | **82.6** | **221.6** | **106.2** |
| *Setting 3 (Cross-Environment)*: | | | | | | | | | | | | |
| MetaFi++ Zhou et al. (2023) | 9.3† | 55.1† | 367.8 | 121.0 | 5.3† | 45.9† | 360.2 | 117.2 | 6.4† | 49.1† | 369.5 | 116.0 |
| HPE-Li D Gian et al. (2025) | 4.3† | 47.8† | 381.1† | 110.3† | 4.2† | 40.3† | 378.2† | 104.0† | 3.4† | 41.9† | 388.4† | 107.9† |
| **DT-Pose (Ours)** | **10.7** | **58.8** | **332.7** | **105.1** | **4.4** | **49.7** | **338.3** | **102.0** | **9.8** | **61.2** | **316.8** | **104.2** |

Table 2: **2D HPE** results and **efficiency comparison** on **WiPose**. Best and second-best are in **Red** and Blue, respectively. ‡ indicates results reproduced from released code, **correcting errors** in D Gian et al. (2025) where MPJPE was mistakenly reported lower than PA-MPJPE. P and F refer to Params and Flops.

| Method | MPJPE↓ | PA-MPJPE↓ | P(M) / F(G)↓ |
|---|---|---|---|
| MetaFi++ | 49.2‡ | 30.1‡ | 25.6 / 502.3 |
| HPE-Li | 40.9‡ | 25.9‡ | **3.5** / 5.2 |
| **DT-Pose (Ours)** | **34.3** | **23.1** | 3.8 / **1.5** |

Table 3: **3D HPE** results on **Person-in-WiFi-3D (1 Person)**. Best and second-best are in **Red** and Blue, respectively. † indicates results reproduced from released code, which are not reported in the original paper.

| Method | MPJPE↓ | PA-MPJPE↓ |
|---|---|---|
| MetaFi++ Zhou et al. (2023) | 132.0† | 75.8† |
| HPE-Li D Gian et al. (2025) | 120.2† | 69.5† |
| Wi-Pose Jiang et al. (2020) | 101.8 | - |
| PiW3D Yan et al. (2024) | 91.7 | - |
| **DT-Pose (Ours)** | **90.0** | **58.7** |

## 4.2 ABLATION STUDY

**Influence of Masking Ratios**. In Fig. 4, the performance improves with higher ratios but drops beyond 80% due to the excessive reconstruction difficulty. Thus, we use the default ratio at 80%.

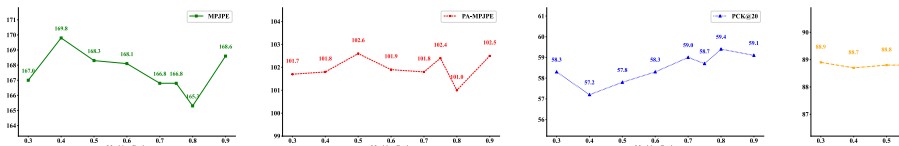

Figure 4: Performance on the **MM-Fi (P1-S1)** with different masking ratios.

**Influence of Masking Strategies**. Table 4(a) shows that the unstructured masking strategy yields the best performance during pretraining, primarily because it captures contextual cues across both time and channel levels.

**Influence of Pre-training Components**. Table 4(b) shows that our pretraining phase learns more general and robust representations than training from scratch with pose supervision. Moreover, the temporal consistency strategy further enhances performance, highlighting the importance of motion-aware learning in WiFi-based HPE.

**Influence of Pose Decoding Components**. Table 4(c) shows that replacing the simple MLP decoder with our adjacent-overarching joint modeling with explicit priors significantly improves absolute

joint localization, underscoring the value of capturing both local and global skeletal dependencies. The task prompt also plays a vital role by adapting skeletal shape information into decoding. If we remove it, the PA-MPJPE metrics will degrade a lot.

Table 4: Ablation studies on the **MM-Fi (P1-S1) dataset**.

(a) **Masking strategies**.

| Masking Strategy | MPJPE↓ | PA-MPJPE↓ |
|---|---|---|
| Channel-Structured | 175.2 | 103.9 |
| Time-Structured | 180.7 | 107.1 |
| **Unstructured (Ours)** | **165.3** | **101.0** |

(b) **Pretraining** component analysis. MO: masked operation. TCS: temporal consistency strategy. [†]: Trained from scratch with pose supervision.

| MO | TCS | Uniformity | MPJPE↓ | PA-MPJPE↓ |
|---|---|---|---|---|
| ✗ | ✗ | ✗ | 198.6[†] | 100.9[†] |
| ✓ | ✗ | ✗ | 183.1 | 102.0 |
| ✓ | ✓ | ✗ | 173.1 | 102.7 |
| ✓ | ✗ | ✓ | 181.8 | 101.9 |
| ✓ | ✓ | ✓ | **165.3** | **101.0** |

(c) **Pose decoder** component analysis. TP: task prompt. GCL: Graph Convolution layers. TL: Transformer layers. [†]: MLPs as the pose decoder; [‡]: we transform all patches into the number of joints by MLPs.

| TP | GCL | TL | MPJPE↓ | PA-MPJPE↓ |
|---|---|---|---|---|
| ✗ | ✗ | ✗ | 197.4[†] | 103.5[†] |
| ✓ | ✗ | ✗ | 174.1 | 101.3 |
| ✗ | ✓ | ✗ | 179.8[‡] | 107.0[‡] |
| ✗ | ✗ | ✓ | 181.4[‡] | 103.0[‡] |
| ✓ | ✓ | ✗ | 166.7 | 103.2 |
| ✓ | ✗ | ✓ | 167.1 | 101.1 |
| ✗ | ✓ | ✓ | 167.0 | 103.3 |
| ✓ | ✓ | ✓ | **165.3** | **101.0** |

### 4.3 QUALITATIVE ANALYSIS

**Temporal Consistency Strategy**. As shown in Fig. 5, incorporating the temporal consistency strategy enhances inter-sequence separability and intra-sequence compactness, thereby strengthening temporal coherence and improving motion discrimination across action sequences.

**Dimension Collapse Phenomenon**. In Fig. 6 (a), we calculate the covariance values of each dimension of the WiFi representations. A more compact distribution is clearly visible when the uniformity term is included, implying that it enriches dimensional diversity and improves inter-dimensional dependencies. Additionally, Fig. 6 (b) represents the singular values of WiFi representations. More large singular values emerge upon introducing the uniformity term, suggesting that the embedding space mitigates the dimension collapse and preserves richer information.

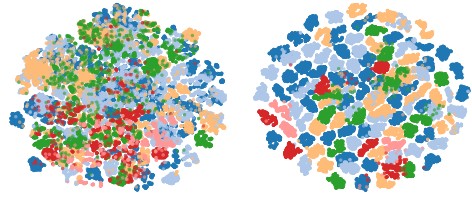

(a) w/o TCS      (b) w/ TCS

Figure 5: t-SNE visualization of WiFi representations on **MM-Fi (P1-S1)** with and without the temporal consistency strategy (TCS) in the pretraining phase. Each color indicates a different action class.

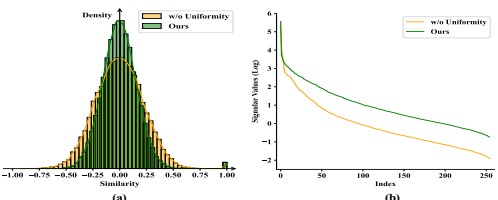

Figure 6: Dimension collapse. (a) represents the statistics of the covariance values of the WiFi representation dimensions. (b) compares the singular values of WiFi representations (zoom in for a better view).

**Masking-Reconstruction Visualization**. In Fig. 7, we plot the raw, masked, and reconstructed WiFi signals, selecting four actions to highlight variations in WiFi signal patterns. Our method faithfully reconstructs the original WiFi signals, underscoring its ability to capture domain-consistent and motion-discriminative WiFi representations effectively.

**WiFi Representations Comparison.** Fig. 8 provides the t-SNE visualization of WiFi representations across multiple models and datasets. In contrast to both the raw WiFi signals and other existing methods, the learned WiFi representations of our proposed method exhibit excellent inter-sequence separability and intra-sequence compactness. Consequently, this motion-discriminative representation space benefits subsequent pose estimation tasks, whereas prior methods tend to learn spurious or motion-irrelevant features that can undermine estimation accuracy.

**Pose Realistic.** Fig. 9 compares predicted poses across various methods and datasets. Our predictions exhibit a more consistent motion tendency in the MM-Fi dataset, highlighted by the green circles. Moreover, as the resolution of WiFi signals increases in the other two datasets, the predicted poses of our method become more coherent and precise. Notably, our predicted skeletal structures adhere closely to the human topology.

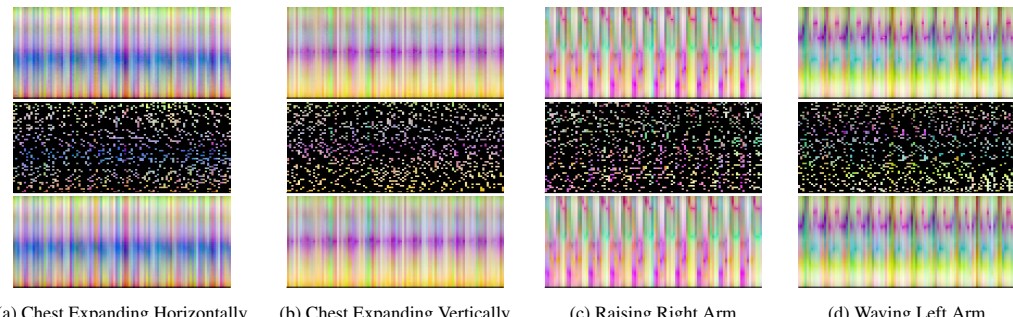

| (a) Chest Expanding Horizontally | (b) Chest Expanding Vertically | (c) Raising Right Arm | (d) Waving Left Arm |

Figure 7: WiFi visualizations on the **MM-Fi (P3-S1)**. The first row represents the raw WiFi signals, the second row represents the masked WiFI input, and the third row denotes the reconstructed WiFi output. All of them contain ten continuous frames.

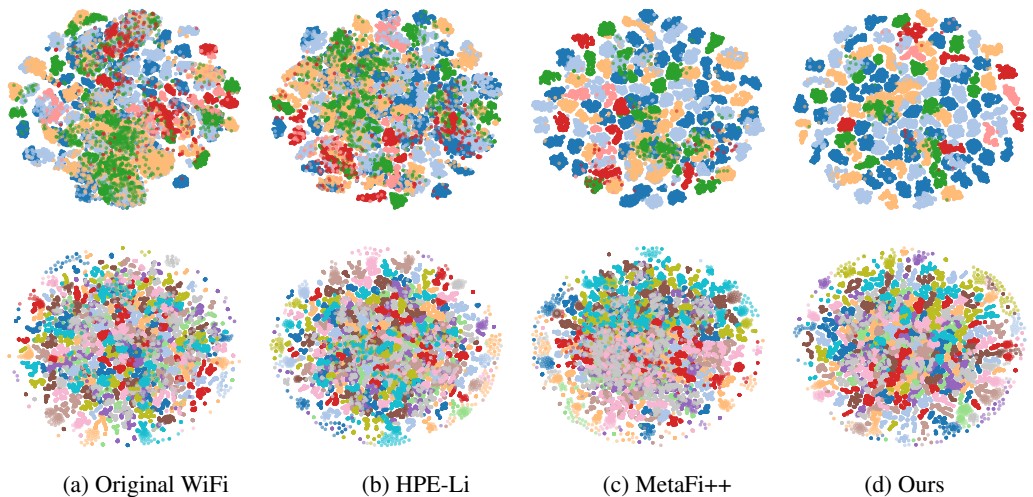

| (a) Original WiFi | (b) HPE-Li | (c) MetaFi++ | (d) Ours |

Figure 8: t-SNE visualization of WiFi representations. The first row denotes the WiFi representations extracted on the **MM-Fi (P1-S1)** testing set, and the second row represents the representations obtained on the **WiPose** testing set. Each color corresponds to a distinct action category.

### 4.4 LIMITATION AND DISCUSSION

**Joints Analysis**. To evaluate the joint-level accuracy of our method, we calculate the pose estimation error for each joint across three different datasets, as shown in Table 5. Our method performs superiorly on coarse-grained body parts like the torso. However, the hands and elbows exhibit the highest errors. These results stem from the limited resolution of current WiFi signals, which hinders the capture of fine-grained actions, e.g., hand movements. In Table 5(b) and 5(c), hand and elbow joints errors decrease when an increased number of antennas and receivers is employed for WiFi signal capture. *Consequently, fine-grained pose estimation with WiFi necessitates higher-resolution signals, whereas images can achieve it even at lower resolutions.*

**Modality Comparison**. Table 6 compares the HPE performance between images and WiFi, highlighting a notable performance gap that can be primarily attributed to two factors:

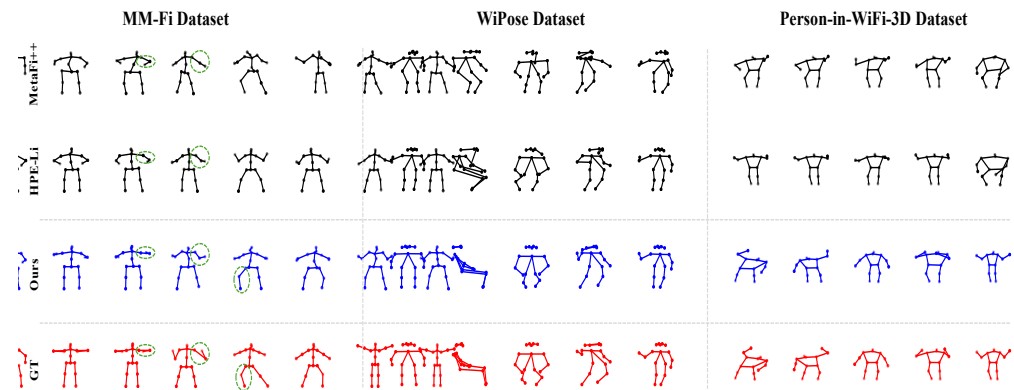

Figure 9: Predicted poses of MetaFi++, HPE-Li, and our method among all datasets.

Table 5: Per-joint performance on three datasets. L. and R. denote left and right, respectively.

(a) **MM-Fi (P1-S1)**

| Joints | MPJPE↓ | PA-MPJPE↓ |
|---|---|---|
| Bot Torso | 102.7 | 56.7 |
| L.Hip | 106.9 | 63.7 |
| L.Knee | 105.7 | 66.1 |
| L.Foot | 104.1 | 88.6 |
| R.Hip | 108.3 | 64.3 |
| R.Knee | 105.3 | 67.4 |
| R.Foot | 109.5 | 90.8 |
| Center Torso | 112.3 | 44.5 |
| Upper Torso | 135.8 | 54.2 |
| Neck Base | 158.2 | 66.2 |
| Center Head | 160.5 | 70.9 |
| R.Shoulder | 147.7 | 73.6 |
| *R.Elbow* | *249.1* | *140.5* |
| *R.Hand* | *364.5* | *284.0* |
| L.Shoulder | 141.8 | 77.2 |
| *L.Elbow* | *235.8* | *132.4* |
| *L.Hand* | *362.4* | *277.0* |
| Average | 165.3 | 101.0 |

(b) **WiPose**

| Joints | MPJPE↓ | PA-MPJPE↓ |
|---|---|---|
| Nose | 30.9 | 14.9 |
| Neck | 27.3 | 11.7 |
| R.Shoulder | 28.7 | 13.4 |
| *R.Elbow* | *38.7* | *25.6* |
| *R.Wrist* | *48.2* | *35.8* |
| L.Shoulder | 29.8 | 16.6 |
| *L.Elbow* | *37.2* | *24.9* |
| *L.Wrist* | *43.3* | *30.6* |
| R.Hip | 24.6 | 17.2 |
| R.Knee | 21.0 | 19.3 |
| R.Ankle | 22.6 | 21.7 |
| L.Hip | 25.6 | 17.9 |
| L.Knee | 22.4 | 19.2 |
| L.Ankle | 26.0 | 22.2 |
| R.Eye | 31.6 | 15.5 |
| L.Eye | 32.4 | 16.3 |
| R.Ear | 30.8 | 14.9 |
| *L.Ear* | *96.8* | *77.7* |
| Average | 34.3 | 23.1 |

(c) **Person-in-WiFi-3D (One Person Setting)**.

| Joints | MPJPE↓ | PA-MPJPE↓ |
|---|---|---|
| Neck | 71.6 | 36.2 |
| Head | 77.6 | 43.1 |
| L.Shoulder | 80.5 | 37.8 |
| R.Shoulder | 81.1 | 37.8 |
| *L.Elbow* | *107.4* | *54.1* |
| L.Hip | 57.7 | 41.5 |
| *R.Elbow* | *114.2* | *55.1* |
| R.Hip | 58.6 | 42.0 |
| *L.Hand* | *164.8* | *117.9* |
| L.Knee | 65.6 | 52.2 |
| *R.Hand* | *179.2* | *122.4* |
| R.Knee | 64.3 | 52.8 |
| L.Ankle | 69.8 | 65.8 |
| R.Ankle | 67.4 | 62.5 |
| Average | 90.0 | 58.7 |

(1) images inherently encode human spatial priors, which are absent in WiFi signals; and (2) the spatial resolution of existing WiFi signals remains limited. *Nevertheless, the two modalities are complementary: WiFi ensures robustness under low-light or occluded conditions, while images provide high-resolution spatial details in well-lit environments.*

Table 6: Modality comparison on **MM-Fi**.

| Modality | MPJPE↓ | PA-MPJPE↓ |
|---|---|---|
| *Protocol 1 - Setting 1 (Random Split)*: | | |
| Image Yang et al. (2024) | 279.0 | **81.2** |
| **WiFi (Ours)** | **178.5** | 104.5 |
| *Protocol 1 - Setting 2 (Cross-Subject)*: | | |
| Image Yang et al. (2024) | 285.3 | **81.9** |
| **WiFi (Ours)** | **221.6** | 106.2 |
| *Protocol 1 - Setting 3 (Cross-Environment)*: | | |
| Image Yang et al. (2024) | **288.6** | **84.1** |
| **WiFi (Ours)** | 316.8 | 104.2 |

## 5 CONCLUSION

In this paper, we revisit and highlight two critical challenges in WiFi-based human pose estimation (HPE): (1) the cross-domain gap and (2) the structural fidelity gap. To tackle these issues, we introduce a two-phase framework, ***DT-Pose***, and evaluate its effectiveness through extensive experiments on both 2D and 3D WiFi-based HPE tasks. Furthermore, we discuss the limitations and advantages of WiFi signals, emphasizing their suitability for Edge AI applications in the AIoT era.

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

## A   OTHER RELATED WORKS

**Masked Pre-training**. Masked pretraining techniques have been widely studied across various data modalities for self-supervised representation learning, leveraging the reconstruction of masked inputs as a core strategy Devlin (2018); Radford (2018); Bao et al. (2021); He et al. (2022); Tong et al. (2022); Wang et al. (2023); Huang et al. (2022); Yan et al. (2023); Cheng et al. (2023). Among these modalities, the BERT Devlin (2018) and GPT Radford (2018) are two seminal language models that pioneered the masked modeling paradigm by predicting masked word tokens based on context information. Inspired by them, the computer vision community introduced masked pertaining frameworks for images, giving rise to representative methods like BEiT Bao et al. (2021) and MAE He et al. (2022), while the video community Tong et al. (2022); Wang et al. (2023) subsequently demonstrated that the masked mechanism extends effectively into the temporal dimension. Beyond these, other data modalities, including audio (Audio-MAE Huang et al. (2022)), skeleton (SkeletonMAE Yan et al. (2023)), and time series (TimeMAE Cheng et al. (2023)), have similarly validated the feasibility and efficacy of masked modeling for task-agnostic representation learning in a self-supervised manner. [*Summary*]: To the best of our knowledge, this work is the first to employ a self-supervised masked pre-training paradigm in the WiFi modality. Moreover, we incorporate the temporal-consistent contrastive strategy with uniformity regularization to extract motion-discriminative representations from sparse and continuous WiFi signals, thereby preserving motion semantic consistency while mitigating potential mode collapse.

**Skeleton-based Action Recognition**. Learning skeleton action representation can be conceptualized as the inverse process of human pose decoding. Typically, skeleton-based action recognition methods can be categorized into CNN-based, GCN-based, and Transformer-based Wang et al. (2016); Chen et al. (2024b; 2021); Song et al. (2022); Chi et al. (2022); Chen et al. (2024a). CNN-based methods transform skeleton sequences into image-like formats to extract discriminative representations Wang et al. (2016); Liu et al. (2017). In contrast, GCN-based methods model human joints and bones as graph nodes and edges, explicitly incorporating a learnable adjacent matrix to explore spatial-temporal features, thus improving performance by a large margin Chen et al. (2021); Song et al. (2022); Chi et al. (2022); Chen et al. (2024c). More recently, Transformer-based methods leverage self-attention mechanisms to capture long-range dependencies among joints Plizzari et al. (2021); Gao et al. (2022); He et al. (2024a). [*Summary*]: Inspired by the success of GCNs and Transformers in learning representations directly from the skeleton data, we combine their strengths in a reverse decoding manner to regress poses from WiFi-based representations. Although WiFi semantic vectors inherently lack human skeletal topology priors, our method compensates for this by incorporating skeletal topology, enabling more realistic and faithful pose predictions.

## B   DATASETS

**MM-Fi** Yang et al. (2024). It comprises 27 distinct action categories performed by 40 volunteers across four different rooms, resulting in approximately 320.76k single-person synchronized frames. One transmitter with one antenna and one receiver with three antennas capture all WiFi signals. Each skeletal pose consists of 17 joints encoded with 3D coordinates. To rigorously assess robustness, the dataset introduces three protocols and three settings for data splitting. Protocol 3 (P3) encompasses all 27 action categories, while Protocol 1 (P1) and Protocol 2 (P2) focus on 14 daily activities and 13 rehabilitation exercises, respectively. Setting 1 (S1 Random Split) randomly divides all data into training and testing sets with a 3:1 ratio. Setting 2 (S2 Cross-Subject Split) employs 32 subjects for training and the remaining 8 subjects for testing. Setting 3 (S3 Cross-Environment Split) selects 3 rooms randomly for training and others for testing.

**WiPose** Zhou et al. (2022). It contains 12 action categories performed by 12 volunteers. WiFi signals are captured by one transmitter with three antennas and one receiver with three antennas. Each pose annotation comprises 18 joints in 2D coordinates. The official split provides 132847 WiFi samples for training and 33753 for testing.

**Person-in-WiFi-3D** Yan et al. (2024). It includes 8 daily actions performed by 7 volunteers at three distinct locations. A single transmitter with one antenna and three receivers with three antennas capture all WiFi signals. Each skeleton pose features 14 joints with 3D coordinates. The dataset has

been officially partitioned into training and test sets, with 89946 WiFi samples allocated for training and 7824 for testing.

## C  IMPLEMENTATION DETAILS

In the pre-training phase, the encoder-decoder is trained for 400 epochs using the AdamW, employing a batch size of 256, a learning rate of 1.5e-4 with cosine annealing schedule, a warm-up epoch of 40, and a weight decay of 0.05. The mask ratio is set as 80%. For the MM-Fi dataset, we train the pose decoder for 50 epochs using the SGD optimizer with a weight decay of 0.01. For the WiPose dataset, we train the pose decoder with the AdamW optimizer for 50 epochs. For the Person-in-WiFi-3D dataset, we train the pose decoder for 200 epochs using the AdamW optimizer. All the learning rates are 1e-3, and the batch size is 32. All the experiments are finished using the PyTorch platform on a GeForce RTX 4090 GPU.

## D  EVALUATION METRIC

Three evaluation metrics are adopted following mainstream methods Yang et al. (2024); D Gian et al. (2025); Zhou et al. (2023):

**Mean Per Joint Position Error (MPJPE (mm))**: Measure the average Euclidean distance between ground truth and predictions, which is widely used to evaluate absolute positional accuracy.

**Procrustes Analysis MPJPE (PA-MPJPE (mm))**: Measure the MPJPE after aligning the predictions and ground truth using rigid transformations (translation, rotation, and scaling) by Procrustes analysis. Typically, it can be used to reflect the similarity in human shape and structure.

**Percentage of Correct Keypoints (PCK@$\alpha$ (%))**:; Measure the percentage of predictions that fall within a certain threshold distance from the ground truth. The threshold is set as a fraction $\alpha$ of the torso length following the previous works D Gian et al. (2025); Zhou et al. (2023). It is widely used to evaluate the local accuracy.

Table 7: **2D HPE** results on **MM-Fi (P3-S1)**. Best and second-best are in **Red** and Blue, respectively.

| Method | PCK@20↑ | PCK@30↑ | PCK@40↑ | PCK@50↑ | MPJPE↓ | PA-MPJPE↓ |
|---|---|---|---|---|---|---|
| Wi-Pose Jiang et al. (2020) | 48.6 | 65.1 | 75.6 | 82.4 | 158.2 | 97.7 |
| Wi-Mose Wang et al. (2021) | 48.7 | 66.6 | 77.3 | 83.9 | 155.8 | 95.4 |
| WiLDAR Deng et al. (2023) | 44.1 | 62.6 | 72.6 | 79.3 | 170.3 | 115.6 |
| WiSPPN Wang et al. (2019a) | 45.4 | 63.2 | 74.1 | 81.0 | 166.5 | 110.0 |
| PerUnet Zhou et al. (2022) | 50.1 | 67.3 | 77.6 | 83.6 | 154.6 | 98.6 |
| MetaFi++ Zhou et al. (2023) | 45.5 | 64.4 | 75.1 | 81.8 | 164.4 | 106.3 |
| HPE-Li D Gian et al. (2025) | 52.1 | 68.2 | 78.2 | 85.1 | 149.4 | 92.5 |
| **DT-Pose (Ours)** | **65.8** | **77.9** | **85.1** | **89.8** | **137.0** | **92.3** |

## E  THE USE OF LLM

The LLM was solely employed as a general-purpose tool to improve writing style after the manuscript had been completed. All polished sentences were carefully reviewed by the authors to ensure the accuracy and integrity of the content.

