# OpenReview forum: "Towards Robust and Realistic Human Pose Estimation via WiFi Signals"
_ICLR.cc/2026/Conference — ICLR 2026 Conference Withdrawn Submission_

### Official Review · Reviewer_2qs6 · 2025-10-28

**Soundness:** 3
**Presentation:** 2
**Contribution:** 2
**Rating:** 4
**Confidence:** 4

**Summary:**

1. The authors propose DP-Pose, a WiFi-based human pose estimation framework. The main components consist of two parts:
(1) a pre-training stage, which includes MAE-based WiFi sample patch embedding training and temporal-consistency training to build an optimized pose encoder;
(2) a pose decoder, which consists of a GCN and a Transformer module that jointly consider human body topology.

2. The MAE-based pre-training stage is quite interesting, as it considers both the intra-sample features within each WiFi signal and the consistency across different WiFi samples. However, its contribution to the declared "cross-domain consistency" remains unclear.

3. The results are promising, but the number of comparison baselines is limited.

4. The ablation study is fairly comprehensive.

5. Several important modules are not well-defined, and their calculation procedures are not described.

**Strengths:**

Refer to the summary part.

**Weaknesses:**

1. The vector representations of the "Temporal Consistency Strategy" part is confusing. My understanding is that e is obtained by pooling embedded patches within a WiFi sample, so how to project e into s and v, the calculation process needs to be explained.

2. The definition and calculation process of “task prompt” need further explanation

3. The overall structure of “Topology-Constrained Pose Estimation” appears to follow a conventional design, which may limit its novelty. Is there a more efficient way to construct joints compared to directly copying the averaged WiFi patch embeddings?

4. The quantitative comparison involves relatively few related methods. Given that many approaches are discussed in the related work section, it is unclear why only two of them were included in the experimental evaluation.

5. In Table 4(b), several modules appear to cause notable variations in MPJPE, whereas P-MPJPE remains relatively stable. Could the authors explain the reason behind this discrepancy?

**Questions:**

Refer to the weakness part.

---

### Official Review · Reviewer_Kq8i · 2025-10-31

**Soundness:** 2
**Presentation:** 2
**Contribution:** 2
**Rating:** 4
**Confidence:** 4

**Summary:**

This paper proposes DT-Pose, a two-stage framework for WiFi-based human pose estimation.
It first learns domain-consistent representations via masked self-supervised pretraining and temporal contrastive learning, then decodes poses with topology-aware constraints using a hybrid GCN–Transformer architecture.
Experiments on several WiFi pose datasets show improved robustness and structural realism compared to prior methods.

**Strengths:**

1. The paper is clearly written and structured.
2. The paper includes comprehensive experiments and well-organized ablation and analysis.

**Weaknesses:**

1. Limited technical novelty. The proposed self-supervised masked pretraining largely follows the MAE paradigm, and the contrastive temporal consistency resembles standard temporal contrastive approaches. The hybrid GCN + Transformer decoder architecture is conceptually similar to prior works in skeleton-based action recognition and pose estimation.

2. Lack of cross-dataset evaluation. Although the paper claims to address cross-domain generalization, all experiments are conducted within each dataset (MM-Fi, WiPose, and Person-in-WiFi-3D) independently. Without cross-dataset experiments (e.g., training on one dataset and testing on another), it remains unclear how well the proposed representation truly generalizes across domains with different datasets.

3. No quantitative metric for temporal consistency. The paper emphasizes the use of a temporal consistency contrastive strategy. Yet, no explicit quantitative metric (e.g., pose velocity smoothness, acceleration continuity, or temporal stability score) is reported to assess the temporal coherence of predicted motion sequences.

**Questions:**

See weaknesses.

---

### Official Review · Reviewer_zxpq · 2025-10-31

**Soundness:** 2
**Presentation:** 2
**Contribution:** 1
**Rating:** 2
**Confidence:** 4

**Summary:**

This paper introduces DT-Pose, a novel two-phase framework for WiFi-based human pose estimation (HPE) that addresses two critical, yet overlooked, challenges: the cross-domain gap and the structural fidelity gap. The first phase, Domain-consistent representation learning, employs a self-supervised masked pretraining paradigm with a temporal consistency contrastive learning strategy to learn robust and motion-discriminative WiFi representations. The second phase, Topology-constrained Pose decoding, uses a hybrid architecture that incorporates explicit skeletal topology priors to produce more realistic and structurally coherent pose predictions. The authors evaluate their method on several benchmark datasets, demonstrating superior performance in both 2D and 3D WiFi-based HPE tasks, especially in cross-domain settings.

**Strengths:**

The primary strength of this paper is its identification and systematic approach to addressing the cross-domain and structural fidelity gaps in WiFi-based HPE. The method is written clearly and facilitates reproducibility. The experimental evaluation is thorough and provides convincing evidence of the superiority of DT-Pose over existing methods.

**Weaknesses:**

There are a few points that could be addressed.
- The novelty of the proposed method is somewhat incremental. How the proposed method provides new insight to ICLR community is not very clear. Masked autoencoder and contrastive learning are well-established techniques in representation learning in images and signals. Structured decoding in human pose estimation is also a known approach (especially widely used in bottom up multi-person pose estimation).
- The evaluation is conducted on datasets with a limited number of subjects and environments. While the cross-domain results are promising, it would be beneficial to see how the method performs on a larger and more diverse dataset.
- The paper focuses on single-person pose estimation. It would be interesting to see if the DT-Pose framework could be extended to the more challenging multi-person scenario.

**Questions:**

1.  The paper explores three different masking strategies (unstructured, channel-structured, and time-structured) and finds that the unstructured approach works best. Do the authors have any intuition as to why this is the case? Does this suggest that the spatial and temporal dimensions of the WiFi signal are more entangled than previously thought?

---

### Official Review · Reviewer_gRj4 · 2025-11-01

**Soundness:** 3
**Presentation:** 3
**Contribution:** 2
**Rating:** 4
**Confidence:** 3

**Summary:**

This paper presents DT-Pose, a two-stage framework for WiFi-based human pose estimation that addresses cross-domain generalization and structural fidelity issues. It combines self-supervised domain-consistent representation learning with topology-constrained pose decoding using GCN and Transformer modules.

**Strengths:**

•  The paper conducts comprehensive evaluations on multiple datasets (MMFi, WiPose, and Person-in-WiFi-3D) and provides thorough ablation studies to validate the effectiveness of each component (e.g., masking types, prompt, GCN/Transformer decoder, uniformity loss).
•  The authors perform t-SNE visualizations, intuitively demonstrating the effectiveness of WiFi signal representations.
•  The paper accurately identifies the cross-domain gap issue in current WiFi-based human pose estimation and leverages the self-supervised MAE framework to alleviate this problem.

**Weaknesses:**

•  The paper lacks an in-depth analysis of why MAE and the temporal consistency loss can alleviate or even resolve the cross-domain gap. In Section 3.4, the statement under “Temporal Consistency Strategy” — “… thereby strengthening temporal coherence and improving motion discrimination across action sequences” — appears logically unsubstantiated. As shown in Figure 5, although the sequence-level representations exhibit inter-sequence separability and intra-sequence compactness (different colors representing different sequences), the representation distribution does not reveal clear class boundaries between different actions, thus failing to demonstrate real motion discriminability.
•  The paper does not sufficiently highlight the advantages of WiFi-based HPE over image-based HPE. Although Section 4.4 (Modality Comparison) claims that “WiFi ensures robustness under low-light or occluded conditions,” this statement lacks supporting quantitative results or visual evidence. Furthermore, Table 6 compares with image-based method（Yang et al.）, which is outdated (this algorithm was proposed in 2019) and not directly comparable, making it unconvincing to claim that WiFi-based and RGB-based HPE achieve comparable performance.

**Questions:**

It is unclear how the positive pairs in the “Temporal Consistency Strategy” are constructed. Specifically, are only \hat{X}_t and \hat{X}_{t+1} treated as positive pairs, or are \hat{X}_{t-1} and \hat{X}_t also considered positives? Moreover, since all frames within the same action sequence belong to the same action class, would treating non-adjacent frames from the same sequence as negative samples confuse the model and contradict the goal of temporal consistency learning?

---

### Note · Authors · 2025-11-13

I have read and agree with the venue's withdrawal policy on behalf of myself and my co-authors.